# Parvovirus RNA Processing: Compact Genomic Organization and Unique Alternative mRNA Processing Mechanisms

**DOI:** 10.3390/v17070984

**Published:** 2025-07-15

**Authors:** Lisa K. Uhl, Olufemi O. Fasina

**Affiliations:** Department of Veterinary Pathology, College of Veterinary Medicine, Iowa State University, Ames, IA 50011, USA; lisauhl@iastate.edu

**Keywords:** parvovirus, mRNA processing, alternative splicing, alternative polyadenylation, alternative translation initiation

## Abstract

Parvoviruses have compact genomic organizations with overlapping open reading frames and thus utilize alternative RNA processing strategies, alternative splicing, alternative polyadenylation, and alternative translation mechanisms to generate a range of diverse proteins encoded within their genome. This comprehensive review provides an update on recent insights into the diverse mRNA processing mechanisms utilized by members of the *Parvoviridae* family, with emphasis on *Bocaparvoviruses* and *Dependoparvoviruses* to expand their protein repertoire and maintain their replicative advantage in infected host cells. It highlights the role of Bocaparvovirus ancillary nonstructural protein NP1; the first parvovirus protein involved in mRNA processing, specifically alternative splicing and alternative polyadenylation.

## 1. Introduction

Parvoviruses are non-enveloped linear single-stranded DNA viruses with palindromic sequences at both ends of the genomes. The palindromic hairpin structures and sequences vary between 120 and 550 nucleotides at the 5′ and 3′ ends. The sequences serve as the origin of replication during genome replication. The genome is generally 4–6 kb in size and packaged in an icosahedral capsid with T = 1, which forms a virion that is about 20–25 nm in diameter. Members of the *Parvoviridae* family are subdivided into three subfamilies: *Parvovirinae*-which represent parvoviruses that infect vertebrate hosts, *Densovirinae*-which encompasses viruses that infect invertebrate hosts, and *Hamaparvovirinae,* which includes viruses like mouse kidney parvovirus and Tilapia parvovirus [1,2,3]. Genera within the Parvovirinae subgroup shown in Figure 1 include *Protoparvovirus, Bocaparvovirus*, *Dependoparvovirus*, *Amdoparvovirus,* and *Erythroparvovirus* [4,5]. The genomes of these genera exhibit diverse structures and organization. *Dependoparvoviruses* and *Erythroparvoviruses* have identical (homotelomeric) inverted terminal repeats at both ends of their genome, while *Bocaparvoviruses*, *Protoparvoviruses*, and *Amdoparvoviruses* have disparate (heterotelomeric) repeats. Homotelomeric parvoviruses package genomes with positive and negative polarity at equal rates, while those with heterotelomeric repeats generally package more negative-strand genomes than positive-strand ones [5].

Parvoviruses have compact genomic organization often divided into two halves, with the 5′ left half encoding the non-structural proteins, while the right half encodes the capsid proteins. Alternative RNA processing strategies- alternative transcription initiation, alternative splicing, alternative polyadenylation, and alternative translation initiation are utilized to maximize the expression of all the proteins encoded within the genome [6]. *Protoparvoviruses* and *Dependoparvoviruses* generate transcripts from two and three promoters, respectively [4,5,6], while a single pre-mRNA, generated from a single promoter in the 5′ end of the genome, is processed into all the transcripts expressed by *Bocaparvoviruses*, *Erythroparvoviruses,* and *Amdoparvoviruses* [7,8,9]. There is one intron in the middle of the genome in all Dependoparvoviruses, while multiple introns are found in the genomes of *Bocaparvoviruses*, *Erythroparvoviruses*, *Protoparvoviruses*, and *Amdoparvoviruses* [6]. Furthermore, the location of polyadenylation *cis*-acting elements varies among all the genera, and all transcripts generated by *Protoparvoviruses* and *Dependoparvoviruses* of non-primate origin are cleaved and polyadenylated at the 3′ end of the genome. In contrast, *Bocaparvoviruses*, *Erythroparvoviruses*, *Amdoparvoviruses*, and *Dependoparvoviruses* of primate origin have polyadenylation sites in the middle of the genome and at the right-hand end of the genome [5]. The entry of parvoviral virions into the cell is mediated by sialic acid and heparan sulfate receptors on the cell surface, and the virus is internalized via receptor-mediated endocytosis into the endosomes in the cytosol [10,11,12]. The phospholipase A2 motif that is conserved in the VP1 unique region of all parvoviral capsid proteins mediates the escape of the genome from the endosome and delivery of the virion into the nucleus [13,14]. The capsid finally uncoats in the nucleus, releasing the genome, which is converted to a double-stranded DNA template for transcription [5,15]. Replication of the genome takes place during the S-phase, and because parvoviruses do not encode any protein that induces the S-phase, replication is totally dependent on host cellular DNA machinery [5,16]. *Dependoparvoviruses* are helper-dependent parvoviruses and replicate efficiently in cells co-infected with helper viruses such as adenoviruses and herpesviruses [17,18]. Conversely, *Bocaparvoviruses*, *Protoparvoviruses*, *Erythroparvoviruses,* and *Amdoparvovirus* are autonomous parvoviruses that replicate independently without helper function once the host cell is in S-phase. Replication of parvovirus genomes takes place in a special compartment in the nucleus called Autonomous Parvovirus Replication bodies (APAR bodies), which are distinctively characterized by the viral genome, viral replication protein NS1/Rep, and recruited cellular DNA replication factors such as PCNA, RFC, cyclin A [16,19]. Parvoviruses reorganize the nuclear environment and recruit host cell proteins for efficient and productive genome replication. Over the past few years, it has been recognized that the host cell response to endogenous and exogenous DNA is co-opted by *Protoparvoarvoviruses*, *Dependoparvoviruses*, *Bocaparvoviruses*, and *Erythroparvoviruses* to modulate parvoviral replication [20,21,22,23].

Parvoviruses are pathogens of great economic importance due to their application in the field of gene therapy and viral oncolytic therapy [24,25]. These applications were designed and developed from results and reports garnered from basic research efforts over many decades, from studies on genome replication, gene expression and RNA processing, capsid structure, and antiviral response to these important small linear single-stranded DNA viruses.

## 2. Parvovirus Replication and Host Interaction

### 2.1. Dependoparvovirus Adeno-Associated Virus

Adeno-associated viruses (AAVs) are *Dependoparvoviruses,* first isolated in 1965 as contaminants of adenovirus-infected cells [26]. However, over the years, about nine serotypes of AAV, namely AAV1-AAV9, have been reported in the literature [27]. AAVs are helper-dependent parvoviruses that rely on the helper functions provided by adenovirus and herpesvirus. Chemical agents that induce breaks in the cell genome, such as hydroxyurea and doxorubicin, can also provide helper functions for efficient replication of AAV in infected host cells [18,28,29]. In the absence of a helper function, the AAV genome is latently integrated into the genome of the infected host cells, usually in chromosome 19 [30,31].

To establish a successful and productive life cycle in the host cell, the AAV virion must enter the cell and transit through the cytoplasm to deliver its genome into the nucleus for amplification, packaging, and egress of new virions. It has been reported that heparan sulfate, sulfate proteoglycans fibroblast growth factor receptor 1 (FGFR1), and sialic acid and AAV receptor (AAVR, also known as KIAA0319L) act as receptors for AAV2 entry [10,12,32,33,34,35]. Similarly, sialic acid and platelet-derived growth factor receptor (PDGF-R) have been shown to participate in the uptake of AAV5 into infected cells [36]. Following receptor binding, the AAV transits through the cytoplasm is mediated by clathrin-coated vesicles, though the endosomal system for delivery into the nucleus, where capsids are uncoated and the genome is released for amplification [37,38].

AAV replicates during S-phase with helper function provided by adenoviral, herpes viral protein products, which can induce co-infected cells into S-phase and modulate the nucleus for the formation of AAV replication compartments. Chemotherapeutic agents such as doxorubicin and ribonucleotide reductase inhibiting agents such as hydroxyurea can also induce a G2/M or S phase arrest that is conducive for AAV replication [39]. Adenovirus helper function is provided by E1A, E2A, E4orf6, E1B55K, and VA RNA. Together, these proteins help to alleviate repression of the AAV2 promoter by transcription factors, degrade inhibitory host factors, and enhance the translation of AAV proteins by shutting down host cell protein translation [17,40,41,42]. Herpes viral proteins such the primase/helicase complex (UL5, UL8 and UL52) and ICP8 are capable of assisting and enhancing AAV replication [43]. Replication of the genome by a “rolling hairpin” mechanism mediated by the inverted terminal repeats at the 5′ and 3′ of the genome [44]. First, the linear ssDNA genome must be converted to a double stranded DNA transcription template through leading strand DNA synthesis initiated from the free 3′-OH at the 3′ITR. This is followed by the replication of the 3′ITR which is mediated from a 3′-OH at the terminal resolution site (TRS) within the ITR created by the endonuclease activity of Rep78. The helicase activity of Rep78 is required to unwind the ITR before Rep78 creates the nick at the TRS. The replication cycle results in formation of various intermediate replicative forms that are finally resolved into linear ssDNA genome with formation of hairpin at ends by refolding of the palindromic sequences. These genomes are packaged into the newly formed capsids through the action of the small replication proteins Rep52 and Rep40 [45]. AAV replicates in special compartment in the nucleus, Rep proteins, viral genome, proteins of helper viruses and host cell proteins such as Replication protein A (RPA), cyclin A, Replication factor 1 (RFC-1) DNA polymerases are recruited to these sites for efficient amplification of the genome [16,46].

### 2.2. Bocaparvovirus Minute Virus of Canines (MVC)

Minute virus of canines (MVC) formerly called Canine parvovirus type 1 was isolated in 1970 from the fecal samples of a healthy dog [47]. MVC is a member of the genus *Bocaparvovirus,* which also includes human Bocaparvovirus 1 (HBoV1)*,* which was isolated in 2005 as the second parvovirus that is associated with disease in humans [48,49]. Bocaparvoviruses are associated with infections of the gastrointestinal and respiratory tract. Experimental studies of MVC in young puppies resulted in anorexia and interstitial pneumonia, which led to respiratory distress and signs of immunosuppression. Infection of dogs during gestation can result in resorption of the embryo, fetal deformities, stillbirth, and abortion, depending on the trimester in which the infection occurred. Serological studies indicate that MVC infection is prevalent worldwide [50]. The pathogenicity and antigenicity of MVC are distinct from Canine parvovirus type 2 (CPV), which belongs to the genus *Parvovirus* and causes severe gastroenteritis in dogs of all ages but has very high fatality rates in unvaccinated young puppies [51].

#### 2.2.1. Bocaparvovirus Minute Virus of Canines (MVC) Genomic Organization and Transcription Profile

The MVC genome is 5.4 kb in size. In contrast to AAV and human B19 virus, it has disparate palindromic sequences at both ends of the genome. The MVC left-end hairpin is 183 nt in length, while the right-hand end is 198 nt in length [52]. This disparate organization of the hairpins is also found in *Protoparvoviruses* and *Amdoparvoviruses*. Due to its hairpins’ heterotelomeric nature, MVC disproportionately encapsulates genomes with 3′ to 5′ orientation during the replication of the MVC genome. About 90% of the MVC virions produced during a single cycle of replication contain a negative-sense linear ssDNA genome [5].

The transcription profile of MVC depicted in Figure 2 shows that it has a single promoter at map unit 6 (P6), which generates a single pre-mRNA that is processed via alternative splicing and polyadenylation to produce 8 transcripts that are translated into the non-structural and structural proteins. The genome can be divided into two halves, with the left half encoding the non-structural protein NS1, while the right half encodes the capsid proteins, VP1 and VP2. Interestingly, MVC and other Bocaparvoviruses encode a unique non-structural protein, NP1, from an open reading frame in the middle of the genome [53]. The amino terminus of NP1 overlaps with the NS1 carboxyl terminus, while the amino terminus of VP1 overlaps with the carboxyl terminus of NP1. However, all the proteins are expressed from different reading frames. NP1 is different from all known parvoviral proteins. MVC NS1 and NP1 are essential proteins required for viral genome replication [52]. MVC has two polyadenylation sites, the proximal polyadenylation site (pA)p and the distal polyadenylation site (pA)d. The proximal polyadenylation site is located in the middle of the genome within an exon that is translated into the capsid protein. This presupposes that MVC (pA)p must be suppressed for transcription read-through into the capsid gene. Aleutian Mink disease virus (AMDV) has a similar proximal polyadenylation site location [9]. The distal polyadenylation site of MVC is situated within the hairpin at the 3′ end of the genome. This is unique and conserved in MVC and Bovine Parvovirus [52].

#### 2.2.2. Bocaparvovirus Minute Virus of Canine (MVC) Replication and Host Cell Interactions

As an autonomous parvovirus, MVC replicates in infected host cells during S-phase without the requirement of helper functions from large DNA viruses or DNA-damaging agents. MVC replicates efficiently and generates new productive virions in Walter Reed canine cell/3873D (WRD), robust replication has also been reported in MDCK cell lines [54]. Parvoviruses generally create a pseudo-S-phase nuclear environment characterized by high concentration of cyclin-dependent kinases and activation of the host DNA damage response (DDR) pathway to facilitate the replication of their genome [55]. This was first reported for autonomous parvovirus Minute Virus of Mice (MVM) and a similar phenomenon has been observed in MVC infection [20,21]. Generally, DNA damage response as an innate antiviral response is exploited by many DNA viruses to manipulate the nuclear environment for a productive life cycle in the host [56]. Though inhibited by some viruses, such as Adenovirus, to prevent concatemerization of their genome, it is co-opted by others to enhance genome replication [57,58,59]. MVC induces an ATM-mediated DNA damage response that facilitates its replication. An intra-S-phase arrest mediated by SMC-1 plays a crucial role in the MVC-induced DDR [60]. Furthermore, MVC infection results in apoptosis of infected cells, characterized by activation of caspases due to viral genome replication. MVC proteins NS1 and NP1 do not induce apoptosis when expressed alone in host cells [61]. Similarly, as reported for H1-parvovirus and minute virus of mice (MVM), the replication of MVC takes place in autonomous Parvovirus replication bodies, generally referred to as APAR bodies [19]. The mechanism exploited by parvoviruses to establish these replicating bodies and recruit host cell factors to these sites is currently unknown. The replication of host cellular DNA is inhibited during MVC replication, as seen in other parvoviruses [62]. MVC genome replication and transcription are modulated by histone modifications. NS1 interacts with cellular histone acetyltransferase KAT5, and there is a significant increase in the expression of KAT5, GTF3C4, and KAT2A during MVC infection. MVC replication results in enhanced H4K12ac acetylation [63].

## 3. Parvoviral mRNA Processing Strategies

Viruses, as obligate intracellular metabolic parasites, require the capacity to orchestrate and modulate the host environment either in the nucleus or cytoplasm for their efficient reproductive life cycle. This warrants the use of a diverse range of proteins expressed from the viral genome with the ability to regulate viral genome replication, transcription and translation of viral protein. Additionally, some are capable of mediating a viral response to host factors that are inhibitory to the virus. Therefore, to achieve these goals, viruses utilize RNA processing strategies to expand their protein repertoire. DNA viruses replicating in the nucleus capitalize on nuclear RNA processing events to utilize the eukaryotic metabolic machinery to generate proteome diversity. These modifications include capping, splicing, and polyadenylation. In addition, the initiation of transcripts from multiple cis-acting elements encoded within the viral genome offers a means of expanding viral protein diversity.

Alternative transcription initiation is a common mechanism in eukaryotic gene expression that generates transcripts of varying lengths within the same gene, utilizing different promoters [64]. This leads to regulated gene expression where an upstream portion of the genome is expressed before the downstream region. Many viruses (HPV [65], adenovirus, herpesviruses [66], adeno-associated viruses AAV [67], minute virus of mice (MVM)) employ this mechanism with their genetic organization such that early genes required to initiate viral transcription and/or genome replication are expressed from an early promoter, and other proteins, especially structural proteins, are expressed later from a downstream promoter usually transactivated by the products of the early genes [6]. In eukaryotic genomes, this can manifest as tissue-specific gene expression based on the response of different promoters to tissue-specific transcription factors [64].

### 3.1. Alternative Splicing

Following its discovery in 1977 from studies of Adenovirus, over the years, splicing has been appreciated as an essential feature of gene expression exploited by eukaryotes and viruses to expand their coding capacity [68]. About 95% of all mammalian genes undergo alternative splicing [69]. Splicing is a crucial gene expression strategy; dysregulation or misregulation of the steps involved in the process has been associated with disease in humans. These include cancer and diseases associated with the nervous system and musculoskeletal system, such as frontotemporal dementia, myotonic dystrophy, and spinal muscular dystrophy (SMA) [70,71]. Recently, splicing has been recognized as a crucial gene expression step required for proper development of hematopoietic cells [72,73]. Splicing decisions are executed by an interplay of many factors that are recruited to specific cis-acting motifs on the transcript. It is a two-step transesterification reaction that ligates the 5′ splice site to the 3′ splice site. The choice of splice site and precision of splicing is modulated by the affinity of the sequences around the splice site to host factors involved in splicing. These sequences, which are referred to as intronic splicing enhancers, exonic splicing enhancers, intron splicing silencers, and exon splicing silencers, are embedded within eukaryotic and viral genomes. These elements recruit splicing factors such as SR proteins and heterogenous nuclear ribonucleoproteins (hnRNPs), and in conjunction with components of the spliceosome—a ribonucleoprotein megaparticle—execute the splicing decisions encoded within the genome [70].

Splicing is intricately associated with transcription, and the C-termini of RNA polymerase II interact with many splicing factors. This helps to load these factors onto the transcripts as the polymerase catalyzes the generation of pre-mRNA. This forms the basis of the recruitment-coupling model of splicing and transcription. Conversely, the kinetic model suggests that the rate of Pol II-mediated elongation affects the outcome of splicing decision on a transcript [74,75]. The generation of diversity via alternative splicing involves various coordinated events, such as cassette exons, mutually exclusive exons, retained intron, competing 5′ splice and competing 3′ splice sites to generate protein isoforms with different structural and functional domains [76].

#### 3.1.1. Bocaparvovirus Alternative Splicing

The *Bocaparvovirus* MVC and HBoV1 are excellent models for studying parvoviral RNA processing. MVC has a single promoter, P6, that initiates pre-mRNA transcription at nt 389 [77]. The pre-mRNA is processed by alternative splicing to generate eight unique mRNA isoforms (R1-R8) that are translated into viral nonstructural proteins (NS-100, NS-84, NS-66, NS-50, NS-40, and NP1) and capsid proteins (VP1 and VP2). There are four introns in the MVC pre-mRNA: 1D (nt 395)/1A (nt 2199), 1D’ (nt 1272)/1A (nt 2199), 2D (nt 2309)/2A (nt 2386), and 3D (nt 2491)/3A (nt 3037) as shown in Figure 2. The unspliced MVC pre-mRNA is translated into the largest NS protein, NS-100. Alternative splicing of the progenitor RNA in the NS region results in the generation of four NS proteins: NS-84, NS-66, NS-50, and NS-40. All NS protein translation is initiated at nt 403 and shares a common N-terminus with the retention of the HUH endonuclease domain in all the proteins. Three NS proteins, NS-84, NS-50, and NS-40, splice the 3D/3A intron with fusion of the NS open reading frame with the NP1 open reading frame and share a unique 20-amino-acid C-terminus with NP1. These proteins lack the classical SF3 helicase case domain, which is present in most parvoviral NS and Rep proteins that modulate viral DNA replication and packaging [78]. The *Bocaparvovirus* MVC 3D/3A alternative splicing event that fused the NS ORF with a different C-terminus ORF is similar to the *Protoparvovirus* MVM splicing event that generates MVM NS2 mRNA, where the NS1 ORF is fused with another ORF at nt 1989. The 3D/3A spliced MVC NS proteins have host–cell dependent function like MVC NS2. MVC mutants that lack the expression of 3D/3A spliced NS proteins are required for replication in the canine WRD cell line and are not required for MVC replication in the human 293T cell line. In like manner, HBoV1 also expresses a host–cell dependent nonstructural protein, NS; however, in contrast to MVC, none of the HBoV1 nonstructural protein mRNAs are generated by HBoV third intron excision. HBoV1 promoter P5, initiates pre-mRNA transcription around nt 291–296. The HBoV1 pre-mRNA has six introns with six splice sites: D1 (nt 337)/A1′ (nt 1016), D1 (nt 337)/A1 (nt 2139), D1 (nt 337)/A2 (nt 2331), D1′ (nt 1212)/A1 (nt 2139), D1′ (nt 1212)/A2 (nt 2331), D2 (nt 2260)/A2 (nt 2331), and D3 (nt 2453)/A3 (nt 3090) as shown in Figure 3.

The *Bocaparvovirus* MVC and HBoV1 third intron excision is modulated and regulated by NP1, the first parvovirus nonstructural protein with a unique role in alternative mRNA processing [77]. Several studies have utilized site-directed mutagenesis and revealed that NP1 termination mutants, NP1 truncation, NP1 missense, and structural-defective mutants in the infectious clone background showed that the excision of the bocaparvovirus intron upstream of the proximal polyadenylation sites is NP1-dependent. NP1 is an intrinsically disordered nucleophoprotein with three repeated terminus SR dipeptides. The MVC SR mutants with alanine substitution showed impaired 3D/3A splicing, suggesting that the domain may be associated with interaction and recruitment of RNA processing factors to enhance NP1-mediated splicing [77,79]. Furthermore, recent studies suggest that MVC NP1 and HBoV1 NP1, similar to order parvovirus ancillary proteins, interact with several cellular RNA-binding proteins involved in splicing, RNA modifications, and DNA replication [80].

The excision of Bocaparvovirus MVC and HBoV1 third intron allows access to the capsid open reading frame and transcription through to the capsid gene. This capsid expression mechanism is different from Dependoparvovirus and Protoparvoviruses that utilize a specific promoter to transactivate the generation of a capsid transcript.

#### 3.1.2. Dependoparvovirus Alternative Splicing

The genome of AAV is about 4.6–4.7 kb in size with identical inverted terminal repeats at the end of the genome. AAV terminal repeats form hairpins, which contain the *cis*-acting elements that recruit the host’s DNA replication factors for viral genome amplification. The hairpins also mediate the packaging of the AAV genome after genome replication, and AAV packages equal amounts of positive and negative strand genomes due to the identity of the hairpin in both genomes. All AAV serotypes have three promoters that generate the transcripts, which are translated into the viral non-structural replication proteins (Rep) and capsid proteins (VP1, VP2, and VP3). The replication proteins expressed from the left half of the genome are classified based on size into large Replication proteins (Rep78 and Rep68) and small Replication proteins (Rep52 and Rep40). There is a central intron in all AAV serotypes, which varies in size; it is 240 nt in the prototype species AAV2, while the AAV5 intron is 322 nt. AAVs have a small, compact genome, which makes them a good intractable model for the study of RNA processing in a viral system [6].

The transcription profile of AAV is subdivided into three groups. First, AAV2, AAV1, AAV3, AAV4, and AAV6 generate transcripts that are polyadenylated at the 3′ end of the genome. The second group includes AAV5 and animal AAVs such as Bovine-AAV and Goat-AAV. The transcripts generated from the two upstream promoters in AAVs are cleaved and polyadenylated at a site within the intron in the middle of the genome. Capsid transcripts are polyadenylated at the 3′ end of the genome, similar to the AAVs in group 1. The third group, which includes avian AAV (A-AAV) [81], combines the features of the first and second groups with 50% of their transcripts undergoing cleavage and polyadenylation at the proximal polyA site in the intron, while the remaining 50% are polyadenylated at the 3′ end of the genome [22].

AAV2, the prototype species in the *Dependoparvovirus* genus, has three promoters, P5, P19, and P40 as shown in Figure 4. The unspliced P5 and P19 transcripts are translated into Rep 78 and Rep 52, while the spliced P5 and P19 transcripts are translated into Rep 52 and Rep 40 [6]. The AAV2 intron has two 3′splice sites, A1 (2201) and A2 (2228). The use of A1 results in the expression of Rep68 and Rep40 major isoforms, while the splicing at A2 results in the expression of the minor isoforms. Capsid transcripts generated from the P40 promoter are spliced at A1 to express VP1, while those spliced at A2 are translated into VP2 and VP3 [6]. Splicing of AAV2 RNA determines the level of Rep68 and Rep40 expression, and it is modulated by Rep78 and the helper function from adenovirus and herpesvirus [82,83,84]. In the absence of Rep78 and helper functions the expression of Rep68 and Rep40 is significantly reduced. AAV2 gene expression is also temporal with gradual accumulation of P5, P19, and P40 generated RNA are infection progresses. This is attributed to the transactivation function of Rep78 generated from the P5 promoter on the downstream P19 and P40 promoters [85]. Similarly, the transcription profile of AAV5 as shown in Figure 5, indicates that AAV5 possesses three promoters, P7, P19, and P41 as seen in AAV2. However, in contrast to AAV2, the RNA generated from P7 and P19 promoters are preferentially polyadenylated in the central intron within the genome [86]. Unspliced P7 and P19 are translated into Rep78 and Rep52. Due to the lack of splicing of the P7 and P19 transcripts, Rep68 is not expressed in AAV5, while Rep40 is expressed from the P19 transcript via alternative translation initiation from an in-frame AUG that is 150 nt downstream of the Rep52 initiating codon [87]. P41-generated RNAs are similarly spliced at the two acceptors of the intron to express VP1 from transcripts spliced at A1 and VP2 and VP3 from transcripts spliced at A2 [6]. Unspliced P41-generated transcripts are detected during AAV5 infection, but their function is unknown. Abundant ITR-generated transcripts are observed in AAV5 compared to AAV2, but the role of this 4.3 kb distally polyadenylated RNA during AAV5 infection is currently unknown [28].

### 3.2. Alternative Polyadenylation

The 3′ end processing of eukaryotic and viral transcripts is a prerequisite that determines the export of the transcripts, stability of the transcripts, and efficiency of translation of the proteins encoded by the transcripts [88]. Polyadenylation is a nuclear gene-expression event whereby multiple adenosine residues are added to the cleaved 3′ end of an adenosine residue. The cleavage and polyadenylation site is defined by the *cis*-acting polyadenylation motif encoded within the genome. This comprises the AAUAAA hexamer and the GU/U rich element, which is usually located 30–45 nucleotides downstream of the hexamer motif. The cleavage site is generally located in the 10–20 nucleotide region between the hexamer and the GU/U-rich element. Polyadenylation is a complex reaction initiated by the recruitment of cleavage and polyadenylation factor (CPSF) and cleavage-stimulating factor (CstF) to the hexamer and GU/U-rich element, respectively; their interaction is stabilized by three other factors, namely, cleavage factor I_m_ (CFI_m_), cleavage factor II_m_ (CFII_m_), and the poly (A) polymerase. Altogether, they modulate an endonucleolytic cleavage at the adenosine at the cleavage site [89]. The choice of a specific polyadenylation site when there are multiple sites at the end of a gene leads to alternative polyadenylation. In recent years, it has become increasingly appreciated that polyadenylation is an essential mRNA processing step modulated by DNA and RNA viruses for an effective, productive life cycle in the host cells.

Alternative 3′ end processing provides another platform during RNA processing for viruses to generate diversity in their proteome and also regulate the expression of their genome [90]. Alternative polyadenylation, which results in the usage of proximal polyadenylation sites, can generate transcripts with 3′UTRs that have reduced binding sites for miRNA and RNA binding proteins [91]. Parvoviruses such as *Bocaparvoviruses*, *Amdoparvoviruses*, and *Dependoparvovirus* of animal origin extensively utilize multiple polyadenylation sites to generate various transcripts for adequate viral gene expression [52,86,92]. Furthermore, viral pathogens such as human papillomavirus (HPV) and human immunodeficiency virus (HIV) utilize alternative polyadenylation to regulate their expression [65,93]. The use of HPV promoter proximal polyadenylation regulates the expression of the virus in differentiating epidermal cells (keratinocytes). The inhibition of cleavage/polyadenylation at the polyadenylation site within HIV 5′LTR mediated by U1snRNP binding ensures transcription readthrough into the *gag* gene [94]. Alternative polyadenylation is a crucial nexus in the regulation of gene expression during many cellular processes [90]. These range from neoplastic proliferation of cells associated with global shortening of 3′UTR, the DNA damage response, thermal stress, IgH class switch during immunoglobulin expression, and T-cell activation [91,95,96].

#### 3.2.1. Dependoparvovirus Alternative Polyadenylation

Parvoviruses have compact genetic organization with single or multiple promoters and overlapping open reading frames. Across different *Parvovirinae* genera, the core polyadenylation cis-acting sequences, associated upstream elements, and downstream elements are organized in unique locations within the genome (Figure 1). Protoparvoviruses have two promoters and a single polyadenylation signal located downstream of the capsid gene on the distal right-hand end. Some parvoviruses have an internal polyadenylation signal within an intron in the middle of the genome (*Dependoparvovirus AAV5*, *Erythroparvovirus B19*) [6]. The polyadenylation of AAV5 P7 and P19 transcripts at the proximal polyadenylation site within the central is a critical and distinctive feature of AAV5. It also exemplifies the fact that AAV5 is an excellent model to study the coupling of splicing and polyadenylation as important RNA processing events that must be efficiently modulated to ensure adequate expression of viral proteins. Regulation of proximal AAV5 polyadenylation site -which lie within in an intron that must be spliced for adequate expression of AAV5 capsid proteins-is mediated by the interaction of U1snRNP to the 5′ splice site of the intron depicted in Figure 6. Binding of U1snRNP to the donor site enhances excision of the intron and inhibition of polyadenylation at the proximal polyadenylation site. This regulation is distance-dependent, such that polyadenylation is inhibited when the promoter is closer and enhanced with increasing distance between the promoter and the polyadenylation *cis*-acting elements [6].

#### 3.2.2. Bocaparvovirus Alternative Polyadenylation

Bocaparvoviruses (Minute virus of canines, MVC, and human bocaparvovirus 1, HBoV1) have a single promoter with an internal core polyadenylation *cis*-acting sequence (pA)p within the capsid exon in the middle of the genome downstream of the capsid-initiating AUG [77].

The bocaparvovirus distal polyadenylation sites (pA)d are located within the right-end palindromic hairpin in MVC and Bovine bocaparvovirus BPV; HBoV has a distal cleavage and a polyadenylation site approximately 400 nt upstream of the right-end hairpin. MVC (GenBank accession number FJ214110.1) has three proximal core polyadenylation (pA)p signals (AATAA, nts 3170–3175, 3228–3233, 3258–3263) with four cleavage sites (3195, 3258, 3279, 3291). These polyadenylation sequences must be suppressed and modulated for transcription readthrough to the distal polyadenylation sites for efficient capsid protein production. This regulation is mediated by a unique and small 20 kDa nonstructural ancillary protein, NP1, expressed from a segment in the middle of the genome downstream of the main NS replication protein [77].

Bocaparvoviral NP1 is the first parvoviral protein shown to modulate alternative polyadenylation. It is an intrinsically disordered, nucleophosphoprotein, RNA-binding protein required for the suppression of capsid mRNA cleavage and polyadenylation at the proximal polyadenylation region that overlaps with the capsid mRNA 5′ end and approximately 160–200 nts downstream of the capsid mRNA-initiating codon at nt 3081. MVC NP1 mutants, including a NP1-structurally defective mutant and several truncated mutants, showed preferential utilization of proximal polyadenylation sites. This NP1-mediated mRNA cleavage and polyadenylation suppression allows transcription readthrough into the capsid gene. Similarly, human bocaparvovirus 1 (HBoV1, GenBank accession number JQ923422) has a similar core proximal polyadenylation (pA)p *cis*-acting sequence organization with five cleavage and polyadenylation sites, located at nt 3295–3300, nt 3329–3334, nt 3409–3414, nt 3440–3445, and nt 3485–3490 identified in HBoV1 mRNA extracted from 293T cells transfected with the infectious clone. NP1 modulates the suppression of three of these sites with canonical AAUAAA hexamer signals. However, only two proximal polyadenylation (pA)p sites, located at nt 3329 and nt 3485, were identified and efficiently utilized in HBoV mRNA extracted from HBoV1-infected human airway epithelium (HAE) culture at an air-liquid interface. The MVC and HBoV1 distal polyadenylation (pA)d core sequences are within the 3′ palindromic hairpin. The MVC distal core AAUAAA polyadenylation sequence is located at nts 5268–5273, with two cleavage and polyadenylation sites at nts 5282 and 5305 (Uhl and Fasina, unpublished). HBoV1 also has one core AAUAAA distal polyadenylation signal at nt 5153–5158 and two distal cleavage polyadenylation sites, (pA)d1 at nt 5171 and a unique (pA)d_REH_ at the right end hairpin. The novel second distal cleavage and polyadenylation site often adds polyA within a wide range across 130 nts from nt 5369 to 5499. The HBoV1 (pA)d_REH_ at the right end hairpin is generally cleaved and polyadenylated at nt 5443 and 5444; this position is adjacent to the right-end stem, which is conserved in MVC and HBoV1. Similarly, MVC nt 5305 is in the same position, and this may suggest a conversed bocaparvoviral polyadenylation and cleavage mechanism at the distal polyadenylation sites in the right end hairpin. The choice of polyadenylation at HBoV (pA)d_REH_ depends on NS1 expression, right-end hairpin, and *cis*-elements of (pA)d [97,98].

The regulation of viral alternative polyadenylation is complex and multifaceted. Viral protein-mediated sequence-specific interaction with the viral and cellular mRNA, polyadenylation complex proteins, and cellular RNA binding proteins has been shown to regulate the choice of cleavage and polyadenylation sites. MVC NP1 binds the viral mRNA between the MVC second intron 3′ splice site and the proximal polyadenylation (pA)p region [99]. MVC NP1 immunoprecipitation and a mass spectrometry assay on WRD cells and 293T cells overexpressing NP1 revealed its interaction with CPSF6 and CPSF5, components of the tetrameric polyadenylation complex, polyadenylation cleavage factor I (CFIm) [79,99]; Figure 7. CPSF5 binds the UGUA motif upstream of the core polyadenylation hexamer sequence. MVC NP1 recruits CPSF5 to the viral mRNA. RNase protection of MVC RNA in 293T with CPSF6 knockdown showed that CPSF6 suppresses cleavage and polyadenylation at the proximal polyadenylation sites [99]. In contrast, CPSF5 knockdown resulted in preferential proximal polyadenylation site usage and inhibition of transcription readthrough to the distal polyadenylation sites [79].

Recently, it was revealed that bocaparvoviral mRNA processing, including the HBoV and MVC proximal polyadenylation sites, can be modulated by DNA methylation [100] and mRNA cytidine N4-acetylation [79], respectively. Inhibition of DNA methyltransferase I (DNMT1) with 5-aza-2′-deoxycytidine (DAC) and shRNA-mediated knockdown of DNMT1 resulted in elevated cleavage and polyadenylation of HBoV1 mRNA at the proximal polyadenylation sites [100]. N-acetyl transferase 10 (NAT10), a writer protein that modulates cytidine N4-acetylation, binds MVC RNA and is required for MVC replication [79]. It also modifies conserved “C-C-G” motifs on the MVC mRNA at several positions (nts 564–588, 1689, 3978, and 3311). MVC genome site-directed mutagenesis revealed that the N4-acetylation at nt 3311 is required for the suppression of proximal polyadenylation and transcription readthrough to the distal polyadenylation sites [79].

**Figure 7 viruses-17-00984-f007:**
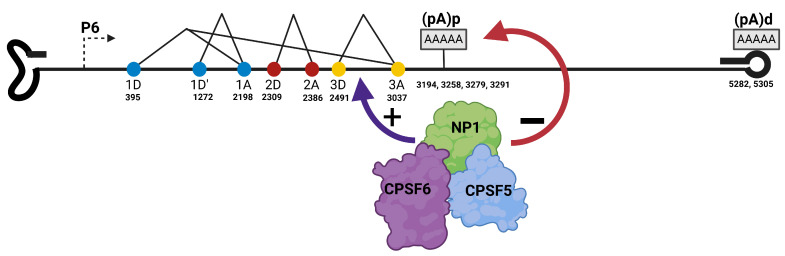
Model of Bocaparvovirus Minute of Canines (MVC) NP1 modulation of alternative polyadenylation and alternative splicing through its interaction with polyadenylation cleavage factor I (CFIm) subunits; cleavage polyadenylation factor 5 (CPSF5) and cleavage polyadenylation factor 6 (CPSF6). The figure was created with Biorender https://BioRender.com.

## 4. Parvovirus RNA Export and Alternative Translation Initiation

While efficient and effective parvovirus mRNA alternative splicing and polyadenylation take place in the nucleus, the transcript must be transported to the cytoplasm for translation and protein production [101,102]. Viral manipulation of the metabolic machinery of the host for diversification of their proteome is not limited to nuclear RNA processing events. Parvoviral mRNA export to the cytoplasm is modulated and regulated by some host cellular proteins. In addition to its role in the MVC third intron splicing and suppression of the proximal polyadenylation region through its interaction with NP1, CPSF6 also affects the export of MVC mRNA [99]. The export of MVC unspliced RNA to the cytoplasm is suppressed by CPSF6, and both MVC transcripts that utilize the proximal polyadenylation and distal polyadenylation sites were exported to the cytoplasm significantly in 293T cells that lacked CPSF6. Similarly, the MVC transcript with retained third intron were exported more than the spliced transcripts. This resulted in increased NP1 expression, a protein translated from the unspliced MVC third intron transcript [99]. It is unclear if HBoV1 mRNA export is modulated by CPSF6; however, it has been shown that HBoV1 NP1 protein nuclear import is modulated by CPSF6, and both proteins are colocalized in the nucleus. Altogether, the reported results indicate that bocaparvoviral RNA export is influenced by CPSF6 for efficient and effective mRNA processing and translation.

Generally, unspliced cellular and viral RNA remain in the nucleus; however, several pieces of evidence have shown that parvoviral unspliced transcripts are exported to the cytoplasm and translated into nonstructural and structural proteins. Unspliced Dependoparvovirus AAV2 transcripts generated from the P5, P19, and P40 promoters have been detected in the cytoplasm [6]. Similarly, the unspliced bocaparvoviral MVC P6-generated transcript is translated to the nonstructural NS-100 protein (Figure 2), and the unspliced HBoV1 P5-generated transcript is translated into NS1-70 (Figure 3). Furthermore, splicing of Dependoparvovirus goose parvovirus influences the choice of translation initiation site in the unspliced P9-generated transcript [103].

Many RNA viruses and some DNA viruses hijack the translation machinery in the cytoplasm for enhanced viral protein production. Translation of viral transcripts is modulated by the cellular ribosomes since no virus encodes its own translation apparatus. The capacity to manipulate translation also affords viruses the opportunity to fine-tune the metabolic state of the host to conditions that are suitable for their genome replication and production of progeny virions. Similarly, modulation of translation by the host is part of the homeostatic responses initiated by the host to evade viral infection [104].

Translation is a complex reaction that involves the integration of many signals from various protein subunits and the ribonucleoprotein megaparticle, ribosome-which modulates the catalytic activities required to decode the information in the transcript and translate it into polypeptides. Translation can be divided into three phases-initiation, elongation, and termination illustrated [105,106]. The initiation of polypeptide production requires the formation of a 43S preinitiation complex, which comprises the 40S ribosome subunits, eukaryotic translation initiation factor I (eIF1), eIF1A, the eIF3 complex, and eIF5, in association with GTP-bound eIF2. The recognition of the cap at the 5′ end of eukaryotic and viral RNAs by eIF4F—a multisubunit complex made up of eIF4E, eIF4A, and eIF4G-coordinates the definition of the 5′ and 3′ ends of the transcript as the poly (A)—binding protein binds to the polyA tail and interacts with eIF4G to activate translation initiation. This interaction confines the recruitment of the 40S ribosome specifically to transcripts with mature 5′ and 3′ termini. Then eIF3 serves as a link between the eIF4F cap recognition complex and the 43S pre-initiation complex. Consequently, the 40S subunit will be positioned on the AUG initiating codon as the pre-initiation complex scans the 5′UTR of the transcript. Then, the 60S subunit joins the 40S to form the 80S subunit, which leads to the release of translation initiation factors and polypeptide chain formation [104].

Many viruses can modulate the translation machinery during any of the phases of this multi-step process. However, the need for viruses to rapidly establish a niche in the host creates conditions for the use of alternative non-canonical translation mechanisms to diversify their protein expression by accessing multiple open reading frames. These mechanisms include internal ribosome entry, ribosome shunting, reinitiation, ribosomal frame-shifting, leaky scanning, non-AUG initiation, and stop-codon readthrough [106]. Generally, these mechanisms are mostly utilized by RNA viruses that replicate and exploit the membranous surfaces in the cytoplasm for genomic amplification and virion production. However, DNA viruses also enhance the protein expression from polycistronic transcripts by capitalizing on some of these non-canonical translation mechanisms. The translation of adenovirus major late capsid protein has been attributed to ribosome shunting [107]. In addition, papillomavirus type 18 expresses E6, E7, and E1 from a polycistronic transcript [108,109]. Furthermore, parvoviruses also hijack the translation machinery for non-canonical translation. Aleutian mink disease virus (AMDV) encodes a tricistronic transcript that is translated into NS2, VP1, and VP2. Similarly, leaky scanning is utilized by AAV2 to express its VP2 and VP3 capsid proteins [110,111,112]. AAV5 P7 transcript is a tricistronic transcript that can be translated into Rep78, Rep52, and Rep40-like protein by alternative translation initiation. The Rep40-like is translated from an AUG that is 150 nt downstream of the Rep52 AUG. While *Dependoparvovirus* AAV2 generates Rep protein diversity by alternative splicing of its internal intron, the AAV5 Rep-encoding P7 and P19-generated RNAs are not spliced because they are polyadenylated at an internal site within the small intron. Although precluded from a splicing-dependent variability, AAV5, as well as all the animal AAVs so far examined, encodes a Rep40-like protein by an internal initiation event from an in-frame AUG 150 nts downstream of the initiating AUG of Rep52. It was shown that in contrast to AAV2, the region between the two small-Rep AUGs in AAV5 contains a signal that is required and sufficient for AAV5 internal initiation, and which can also program both the internal initiation of the AAV2 internal AUG and the AUGs of other heterologous genes.

Although the AAV5 Rep40-like protein contains the three Walker motifs characteristic of the AAV2 Rep proteins and of other SF3-helicases, it is otherwise quite different from AAV2 Rep40, varying at both the amino and carboxyl termini. The AAV2 Rep40 protein is bimodular with a small helical bundle at the amino terminus and a large α/β domain at the C-terminus, while the AAV5 Rep40-like protein lacks this helical bundle at the amino terminus [113]. The AAV5 Rep40-like protein retains significant helicase activity and it will be interesting in the future to compare the biochemical properties and functions of AAV2 and AAV5 small Rep proteins.

## 5. Conclusions

Viruses, as obligate intracellular metabolic parasites, require the capacity to orchestrate and modulate the host environment either in the nucleus or cytoplasm for their efficient reproductive life cycle. This warrants the use of a diverse range of proteins expressed from the viral genome with the ability to regulate viral genome replication, transcription, and translation, in addition to antagonizing host factors inhibitory to the virus. Therefore, in order to achieve these goals, viruses utilize RNA processing strategies to expand their protein repertoire. RNA processing events such as transcription initiation, capping, splicing, and 3′-end processing afford viruses the opportunity to utilize the eukaryotic metabolic machineries for generating proteome diversity. Parvoviruses and other DNA viruses effectively capitalize on their use of nuclear eukaryotic metabolic machineries to co-opt host cell factors for optimal replication and gene expression. Parvoviruses with small genome size and overlapping open reading frames utilize alternative transcription initiation, alternative splicing, and alternative polyadenylation to coordinate the expression of their non-structural and structural proteins. This review comprehensively documents parvovirus MRNA processing with emphasis on the unique mRNA processing strategies of Bocaparvoviruses: Minute virus of canines (MVC), Human Bocaparvovirus 1 (HBoV1), and Dependoparvoviruses (AAV5 and AAV2). While Bocaparvoviruses generate a single pre-mRNA from a single promoter that is processed to multiple transcripts by alternative splicing and alternative polyadenylation, Dependoparvoviruses transactivate multiple promoters to generate multiple transcripts. These transcription strategies also regulate the switch between the early expression of parvoviral nonstructural proteins and late capsid protein expression for efficient viral genome packaging. Bocaparvoviruses encode a unique nonstructural protein NP1, required for the splicing of the intron immediately upstream MVC proximal polyadenylation region. This modulation reveals that in addition to its role in mediating transcription readthrough across MVC (pA)p, NP1 also modulates alternative splicing of MVC pre-mRNA. It was shown that HPV E2, an essential protein required for HPV genome replication, also mediates the readthrough of HPV transcription across its early polyadenylation site for enhanced late gene expression via its interaction with CPSF30, a subunit of the polyadenylation machinery [114]. MVC NP1 interacts with components of the tetrameric polyadenylation complex, cleavage factor I (CFIm), CPSF5, and CPSF6 [79,99]. The transcription profile of MVC transcripts from WRD cells transfected with NP1 mutants, CPSF5 mutant, and CPSF6 mutant suggests a complex multifactorial mechanism in the effective suppression of cleavage and polyadenylation at the MVC proximal polyadenylation and excision of the third intron. CFIm functions as an SR-protein-like enhancer and modulator of polyadenylation and splicing through its interaction with the polyadenylation cis-acting sequences and splicing cis-acting sequences [115,116,117]. The overall model suggests that NP1, its CFIm interaction, may regulate the access and binding of polyadenylation and splicing factors around the sequences in the middle of the genome (Figure 7). While *Dependoviruses* and *Parvoviruses* transactivate downstream promoters to gain access to the capsid gene, Bocaparvoviruses may have evolved to utilize a unique small nuclear NP1 protein to modulate their RNA processing for efficient capsid expression.

## Figures and Tables

**Figure 1 viruses-17-00984-f001:**
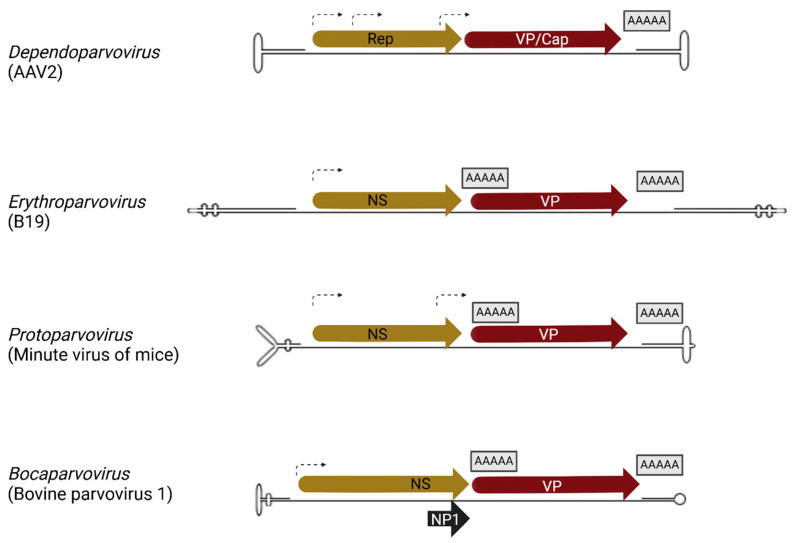
Overview of the *Parvovirinae* subfamily illustrating its genera. The genetic organization of each genus is shown with the genomes as horizontal lines terminating in the palindromic hairpin at both ends. The left half and right half open reading frames are depicted with light green and dark brown arrowed boxes, respectively. The promoters are represented by broken arrows, and the polyadenylation cis-acting elements by the AAAAA sequence block. The figure was created with Biorender https://BioRender.com.

**Figure 2 viruses-17-00984-f002:**
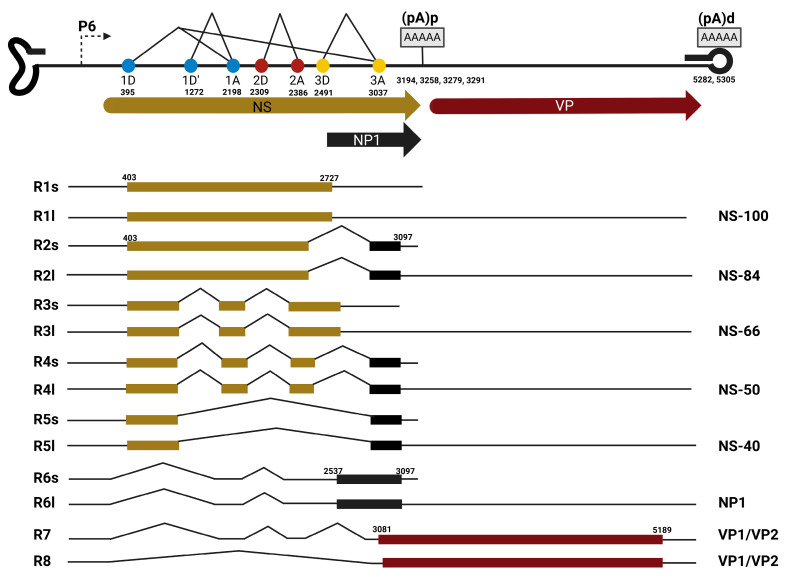
Minute Virus of Canines (MVC) Genomic Organization and Transcription Map: The transcription profile of Minute Virus of Canines (MVC) is shown, depicting the promoter (P6) and the transcripts generated from the single pre-mRNA initiated from the promoter. The viral proteins encoded within each transcript and their corresponding proteins are shown on the right. The figure was created with Biorender https://BioRender.com.

**Figure 3 viruses-17-00984-f003:**
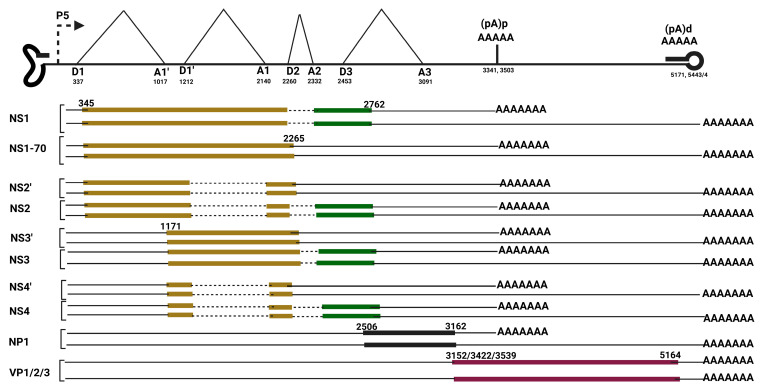
Human Bocaparvovirus 1 (HBoV1) Genomic Organization and Transcription Map: The transcription profile depicts the promoter (P5) and the transcripts generated from the single pre-mRNA initiated from the promoter. The mRNA processing cis-acting sites: 5′ splice sites (D1, D1′, D2, and D3), 3′ splice sites (A1′ A1, A2, and A3), proximal polyadenylation site (pA)p, and distal polyadenylation site (pA)d sites are indicated. The viral proteins encoded within each transcript are shown on the left. The figure was created with Biorender https://BioRender.com.

**Figure 4 viruses-17-00984-f004:**
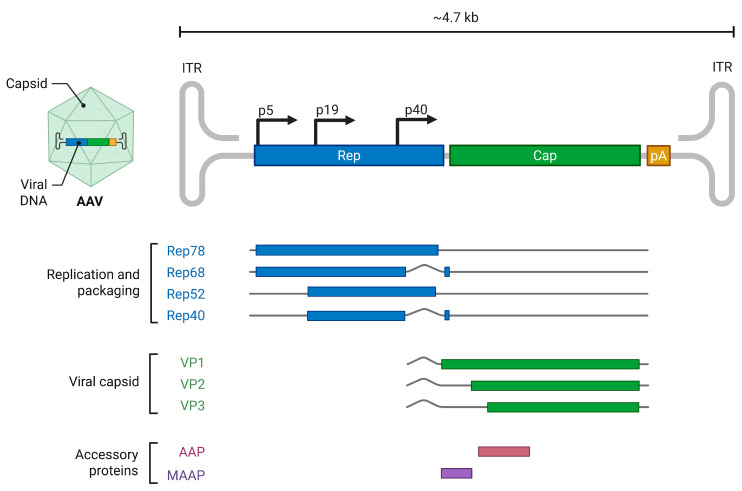
Adeno-associated virus serotype 2 (AAV2) Genomic Organization and Transcription Map: The transcription profile depicts the promoters (P5, P19, and P40) and the transcripts generated from the pre-mRNA initiated from the promoters. The replication and packaging proteins Rep78, Rep68, Rep52, and Rep40 transcripts are depicted in blue. The capsid proteins (VP1, VP2, and VP3) are shown in green, while the accessory proteins, assembly activating protein (AAP) and membrane-associated accessory protein (MAAP) proteins are depicted in brown and purple bars. The viral proteins encoded within each transcript are shown on the left. The figure was created with Biorender https://BioRender.com.

**Figure 5 viruses-17-00984-f005:**
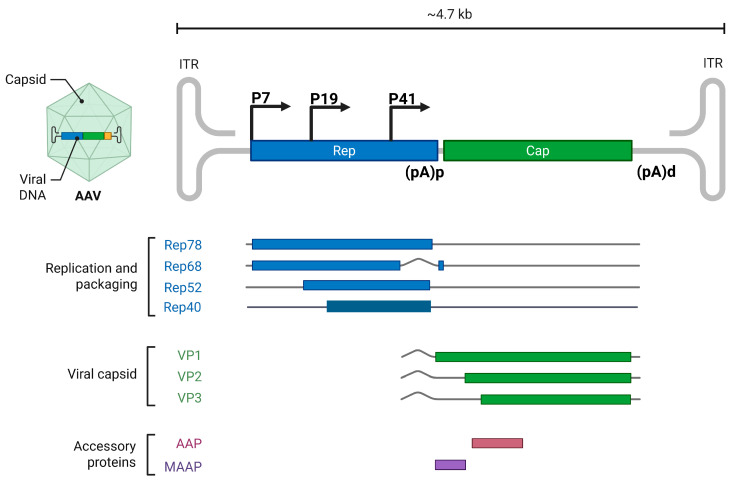
Adeno-associated virus serotype 2 (AAV5) Genomic Organization and Transcription Map: The transcription profile depicts the promoters (P7, P19, and P41) and the transcripts generated from the pre-mRNA initiated from the promoters. The replication and packaging proteins Rep78, Rep68, Rep52, and Rep40 transcripts are depicted in blue. The capsid proteins (VP1, VP2, and VP3) are shown in green, while the accessory proteins, assembly activating protein (AAP) and membrane-associated accessory protein (MAAP) proteins are depicted in brown and purple bars. The viral proteins encoded within each transcript are shown on the left. The proximal polyadenylation site in the middle of the genome within the intron and the distal polyadenylation site at the right end of the genome are depicted. The figure was created with Biorender https://BioRender.com.

**Figure 6 viruses-17-00984-f006:**
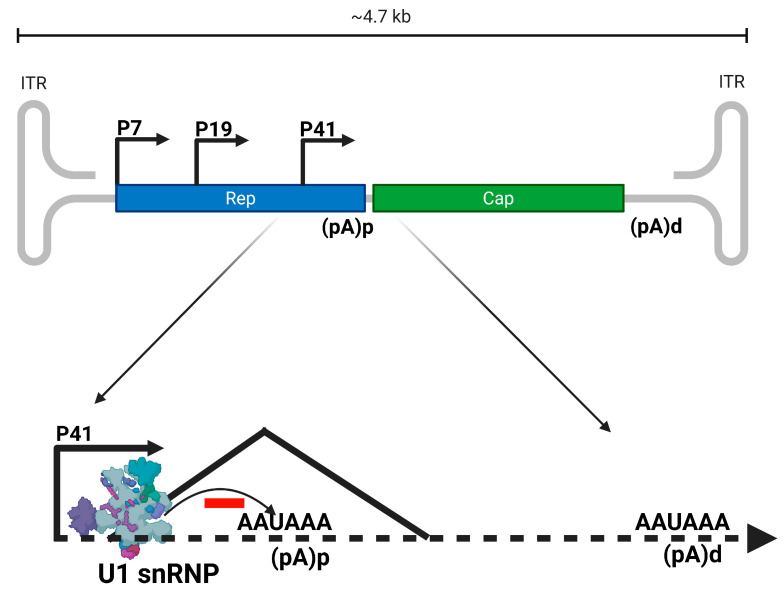
Model of Dependoparvovirus Adeno-associated virus serotype 5 (AAV5) U1 snRNP-mediated distance-dependent alternative polyadenylation regulation. The recruitment of U1 snRNP to the 5′ splice site inhibits P41-generated transcripts’ 3′ end cleavage and polyadenylation at the proximal adenylation site in the middle of the genome. The figure was created with Biorender https://BioRender.com.

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
