# Peer review of "Parvovirus RNA Processing: Compact Genomic Organization and Unique Alternative mRNA Processing Mechanisms"

_viruses, 2025, doi:10.3390/v17070984_

Round 1
Reviewer 1 Report
Comments and Suggestions for Authors
Parvoviruses have a compact genomic organization, generating diverse proteins through overlapping open reading frames and utilizing alternative RNA processing strategies, such as alternative splicing, alternative polyadenylation, and alternative translation mechanisms. This review focuses on the mRNA processing mechanisms of Bocaparvovirus and Dependoparvovirus in the family Parvoviridae, highlighting the role of the Bocaparvovirus auxiliary nonstructural protein NP1 in mRNA processing, particularly in alternative splicing and alternative polyadenylation.The comments on this paper are as follows
1 In the latest virus classification, Parvoviridae includes three subfamilies, namely Densovirinae, Hamaparvovirinae, and Parvovirinae. Parvovirinae contains 11 genera.Please refer to the latest virus classification report.https://ictv.global/taxonomy
2 There is inconsistency in title formatting: some titles are entirely in uppercase words, while others are not.
3 The text includes "5. Conclusions" and "1. Introduction," but lacks sections 2, 3, and 4. The titles are chaotic and difficult to follow.
Author Response
Comments 1: There is no real discussion of the unique challenges for parvoviruses with such small genomes. What is unique about them and their need for manipulating RNA processing? Conversely, what have we learned from studying this that could have implications in other systems?
Response 1: Thanks for the suggestions, we addressed the unique parvovirus genomic organization, overlapping ORFs, and challenges associated with their small genomes in the introduction and extensively discussed them in the section on alternative splicing and polyadenylation.
Comments 2: The figures are all pretty similar and just give transcription maps. A more thoughtful figure could compare between viruses and look for common features that they share.
Response 2: The bocaparvovirus transcription maps have similar features with differences in splicing patterns and open reading frames termination codon position.
In order to provide more context and highlight different parvovirus transcription and RNA processing strategies, we have added two new figures (fig. 4 and fig. 5): AAV2 and AAV5 transcription prolife to the Dependoparvovirus alternative splicing and polyadenylation section. Furthermore, we added an additional figure (fig.6) to highlight and explain the unique AAV5 distance-dependent U1 snRNP mediated alternative polyadenylation regulation.
Comments 3: I think there could be more discussion of the roles of cellular proteins in the mechanisms and this could be incorporated more into the figures.
Response 3: We have added additional information and context to the role of polyadenylation factor CPSF6 in RNA export, alternative splicing and alternative polyadenylation.
Similarly, we provided information on MVC genome replication and its interaction with cellular proteins, including histone acetyltransferase.
Comments 4: There is no discussion of whether this manipulation of mRNA processing is just for viral genomes and what happens to cellular transcripts in the context of parvovirus infection.
Response 4: Parvoviral mRNA processing has been extensively studied and reported with emphasis on viral transcript processing. Parvoviruses do not induce cellular mRNA shutdown; however, whether the interaction of NP1 with cellular polyadenylation factor CPSF6 and CPSF5 affects cellular RNA metabolism is still unclear.
Comments 5: There is minimal discussion of RNA stability and transport which are important for the outcome of mRNA processing on protein translation.
Response 5: Thanks for the suggestion, we have provided additional information on the parvoviral RNA export in the context of how nuclear RNA processing events modulate parvoviral mRNA export to the cytoplasm.
Minor:
Comments 6: Line 304 refers to Fig. 1-8 but it is unclear which figure they are referencing in this article. The same is true on line 390 for Fig. 1-3, and line 411 for Fig.1-4, and line 419 for Fig.1-9. Are these lines from a different article since they refer to figures in a different way?
Response 6: We have addressed the figure annotation with appropriate titles and legends
Comments 7: Figure 3 lacks a title and legend. Is this MVC or HBoV1?
Response 7: We have addressed the figure annotation with appropriate titles and legends.
Reviewer 2 Report
Comments and Suggestions for Authors
The small size and compact organization of the parvovirus genome necessitates ways to maximize gene expression through overlapping ORFs, and alternative RNA processing strategies. This review provides a comprehensive overview of the diverse mRNA processing mechanisms used by different parvoviruses, mainly focusing on the bocaparvoviruses and dependoparvoviruses. The review introduces parvoviruses, describes their replication, and then focuses on their mRNA processing strategies. They describe in detail alternative splicing and alternative polyadenylation, and then more briefly comment on translation. The review is timely and covers the topic well. I have a few suggestions for how it might be broader and of wider appeal.
Suggestions:
- There is no real discussion of the unique challenges for parvoviruses with such small genomes. What is unique about them and their need for manipulating RNA processing? Conversely, what have we learned from studying this that could have implications in other systems?
- The figures are all pretty similar and just give transcription maps. A more thoughtful figure could compare between viruses and look for common features that they share.
- I think there could be more discussion of the roles of cellular proteins in the mechanisms and this could be incorporated more into the figures.
- There is no discussion of whether this manipulation of mRNA processing is just for viral genomes and what happens to cellular transcripts in the context of parvovirus infection.
- There is minimal discussion of RNA stability and transport which are important for the outcome of mRNA processing on protein translation.
Minor:
- Line 304 refers to Fig. 1-8 but it is unclear which figure they are referencing in this article. The same is true on line 390 for Fig. 1-3, and line 411 for Fig.1-4, and line 419 for Fig.1-9. Are these lines from a different article since they refer to figures in a different way?
- Figure 3 lacks a title and legend. Is this MVC or HBoV1?
Author Response
Comments 1: In the latest virus classification, Parvoviridae includes three subfamilies, namely Densovirinae, Hamaparvovirinae, and Parvovirinae. Parvovirinae contains 11 genera.Please refer to the latest virus classification report.https://ictv.global/taxonomy
Response 1: We appreciate the comments and suggestions. The Parvovirinae classification has been updated in the appropriate section with adequate references in the revised manuscript.
Comments 2: There is inconsistency in title formatting: some titles are entirely in uppercase words, while others are not.
Response 2: We have addressed the title formatting in the revised manuscript.
Comments 3: The text includes "5. Conclusions" and "1. Introduction," but lacks sections 2, 3, and 4. The titles are chaotic and difficult to follow.
Response 3: We have addressed the section formatting and order in the revised manuscript.
Round 2
Reviewer 1 Report
Comments and Suggestions for Authors
This review focuses on parvovirus RNA processing, highlighting their compact genomes and use of alternative mRNA processing mechanisms like splicing, polyadenylation, and translation to generate diverse proteins. The review also covers parvovirus replication, host interactions, RNA export, and alternative translation initiation. These mechanisms help parvoviruses expand their proteome and maintain replicative advantage, with implications for gene therapy and oncolytic therapy.